# Cannabidiol Suppresses Angiogenesis and Stemness of Breast Cancer Cells by Downregulation of Hypoxia-Inducible Factors-1α

**DOI:** 10.3390/cancers13225667

**Published:** 2021-11-12

**Authors:** Min Jee Jo, Bu Gyeom Kim, Woo Young Kim, Dae-Hee Lee, Hye Kyeong Yun, Soyeon Jeong, Seong Hye Park, Bo Ram Kim, Jung Lim Kim, Dae Yeong Kim, Sun Il Lee, Sang Cheul Oh

**Affiliations:** 1Graduate School of Medicine, College of Medicine, Korea University, Seoul 08308, Korea; minjeeyoyo@naver.com (M.J.J.); qnrua10047@naver.com (B.G.K.); katecoco@hanmail.net (H.K.Y.); psh3938@hanmail.net (S.H.P.); derrickdyblue22@gmail.com (D.Y.K.); sachoh@korea.ac.kr (S.C.O.); 2Department of Surgery, Korea University Guro Hospital, Korea University College of Medicine, Seoul 08308, Korea; silee@korea.ac.kr; 3Department of Marine Food Science and Technology, Gangneung-Wonju National University, Gangwon 25457, Korea; 4Department of Oncology, Korea University Guro Hospital, Korea University College of Medicine, Seoul 08308, Korea; jensyj85@gmail.com (S.J.); ilovewish777@naver.com (B.R.K.); clickkjl@naver.com (J.L.K.)

**Keywords:** cannabidiol, HIF-1α, angiogenesis, stemness, breast cancer

## Abstract

**Simple Summary:**

Cannabidiol (CBD), one of the compounds present in the marijuana plant, has antitumor properties. However, the effect of CBD on breast cancer remains unclear. The aim of this study was to assess the effects of CBD for the angiogenesis and stemness of breast cancer cells by decreasing the expression of hypoxia-induced factor-1α (HIF-1α) through the Src/von Hippel–Lindau tumor suppressor protein (VHL) interaction. CBD can suppress angiogenesis and stem cell-like properties of breast cancer through Src/VHL/HIF-1α signaling.

**Abstract:**

To assess the effect of Cannabidiol (CBD) on the angiogenesis and stemness of breast cancer cells as well as proliferation. Methods: mRNA level and the amount of protein of vascular endothelial growth factor (VEGF) were determined by qRT-PCR and ELISA. The angiogenic potential of breast cancer cells under hypoxic conditions was identified by the HUVEC tube formation assay. The degradation of HIF-1α by CBD and the Src/von Hippel–Lindau tumor suppressor protein (VHL) interaction were assessed by a co-immunoprecipitation assay and Western blotting. To identify the stemness of mamospheres, they were evaluated by the sphere-forming assay and flow cytometry. Results: CBD can suppress angiogenesis and stem cell-like properties of breast cancer through Src/VHL/HIF-1α signaling. CBD may potentially be utilized in the treatment of refractory or recurrent breast cancer.

## 1. Introduction

Cannabidiol (CBD) is a non-psychoactive constituent of phytocannabinoids in industrially used Cannabis sativa [1], which has been studied for its antineoplastic actions, shifting sights towards a new scope of anticancer treatments [2]. CBD has been reported to inhibit angiogenesis related to the proliferation of estrogen receptor-positive breast cancer [3] by inactivating proangiogenic factors such as vascular endothelial growth factor (VEGF), integins or angiopoietins, matrix metalloproteinase-2/9 (MMP-2/9), the urokinase-type plasminogen activator (uPA), endothelin-1 (ET-1), and platelet-derived growth factor-AA (PDGF-AA), or by activating inhibitory effectors such as thrombospondins or interferons [4,5,6]. Furthermore, CBD induces autophagy and cell death under oxidative stress conditions in cancer and results in impaired endothelial cell proliferation, migration, and tumor microvasculture by inhibiting protein kinase B (Akt), ERK, PI3K, mammalian target of rapamycin (mTOR) signaling, adenylate cyclase, J-Jun, NF-kB activity, actin stress fibers and focal adhesion formation, and FAK and JNK signaling [4,5,6,7,8]. In addition, CBD reduces the secretion of cytokines such as the granulocyte-macrophage colony-stimulating factor from cancer cells. Consequently, reduced recruitment of macrophages from the tumor microenvironment by decreasing the secretion of cytokines suppresses angiogenesis [9]. Thus, CBD inhibits angiogenesis via interference with pro-angiogenic pathways/signaling, factors, and tumor microenvironments. Moreover, CBD induces the interplay between apoptotic and autophagy-inducing signals, resulting in a non-receptor-mediated programmed cell death of breast cancer [3], whereas it has shown little effect on non-tumorigenic mammary cells [2]. Intratumoral hypoxia is one of the most important tumor microenvironments and causes treatment failure due to reactive oxygen species and DNA damage [10,11]. Hypoxia is also required for maintaining the undifferentiated status of stem/precursor cells [12]. It has been reported that breast cancer stem cells (BCSCs) mediate the recurrence or metastasis of breast cancer, which is in line with the consensus that cancer stem cells are the driving force of cancer evolution and resistance to therapies [13]. This impact of hypoxia is mediated by hypoxia-inducible factors-1α (HIF-1α), which is known to serve as s master regulator of oxygen homeostasis and play key roles in the angiogenesis, tumor invasion, and metastasis of breast cancer cells, as well as the maintenance, survival, and expansion of BCSCs [12,14]. HIF-1α induced angiogenesis in hypoxic tissues may be mediated by the activation of VEGF pathway [15]. CBD inhibition of HIF-1α and VEGF in glioma cells represented its anti-neoplastic and anti-angiogenic aspects [16]. HIF-1α in BCSCs also leads to the activation of pluripotency factors such as *Nanog, Oct4*, and *Sox2* in response to hypoxia or cytotoxic chemotherapy [17]. However, the putative anti-tumoral CBD effect on angiogenesis through HIF-1α signaling in breast cancer or BCSCs is poorly understood. Thus, we aimed to investigate how CBD can interfere with HIF-1α signaling, followed by inhibition of the proliferation and angiogenesis of breast cancer.

## 2. Methods

### 2.1. Cell Culture

Human breast cancer (MCF7, MDA-MB-231, T47D, and SK-BR-3) cells were purchased from the Korean Cell Line Bank (Seoul, Korea). Normal human breast (MCF10A) cells and primary human umbilical vein endothelial cells (HUVECs) were purchased from the American Type Culture Collection (ATCC, Manassas, VA, USA). MCF7, T47D, and SK-BR-3 cells were grown in the RPMI-1640 medium (Gibco, Grand Island, NY, USA), and MDA-MB-231 cells were grown in Dulbecco’s modified Eagle’s medium (DMEM; Gibco) containing 10% fetal bovine serum (FBS, Sigma, Darmstadt, Germany) with 100 mg/mL penicillin and streptomycin (P/S, GenDEPOT, Barker, TX, USA). MCF10A cells were grown in the mammary epithelial basal medium (CC-3151; Lonza, Basel, Switzerland) containing supplements (CC-4136; Lonza). HUVECs were grown in endothelial basal medium-2 (CC-3156; Lonza) containing supplements (CC-4176; Lonza) according to the manufacturer’s instructions. 

### 2.2. Generation of Sphere-Forming Cells

Cells were cultured in ultra-low-coated dishes for two weeks in DMEM/Ham’s F12 medium (Gibco) containing 20 ng/mL human epidermal growth factor (hEGF), 10 ng/mL human fibroblast growth factor (hFGF), B-27 supplement, and antibiotics. Fresh DMEM/Ham’s F12 medium was added to each dish (1 mL/dish) every day. After 2 weeks, sphere-forming cells were resuspended in TryPLE (Gibco). Resuspended single cells were seeded in dishes with or without CBD treatment. Then, sphere-forming cells were examined using a light microscope, and the diameter of the spheroids in the dishes was measured. 

### 2.3. Reagents and Antibodies

CBD was obtained from Sigma-Aldrich (St. Louis, MO, USA). CBD dissolved in absolute ethanol (EtOH) was stored at −20 °C. Antibodies against HIF-1α, E-cadherin, and N-cadherin were purchased from BD Biosciences (San Jose, CA, USA). Anti-Snai1, Src, VHL, ubiquitin, CD133, ALDH1A1, and Nanog were purchased from Santa Cruz Biotechnology (Dallas, TX, USA). Anti-Slug, Snai1, and Vimentin were purchased from Cell Signaling Technology (Danvers, MA, USA). Anti-SOX2 was purchased from R&D Systems (Minneapolis, MN, USA). The anti-β-actin antibody was purchased from Sigma-Aldrich. The secondary antibodies anti-mouse-IgG-horseradish peroxidase (HRP) and anti-rabbit-IgG-HRP were purchased from Cell Signaling Technology. Anti-CD24 and CD44 antibodies were purchased from Invitrogen (Carlsbad, CA, USA). Cobalt (II) chloride (CoCl₂) and MG132 (proteasome inhibitor) were purchased from Sigma Aldrich.

### 2.4. Cell Proliferation Assay

Cell proliferation was assessed by WST assay using the EZ-CyTox cell viability, proliferation, and cytotoxicity assay kit (DoGEN, Daeil Lab Service Co. Ltd., Seoul, South Korea). Cells were seeded at a density of 1.2 × 10^4^ cells per well in 96-well plates. Cells were treated with CBD for 24 h and then treated with 10 µL of the WST-1 solution for 4 h at 37 °C. Absorbance was measured at 450 nm using a microplate reader (SPECTRA190; Molecular Devices, Sunnydale, CA, USA).

### 2.5. Immunoblotting

Cells were lysed in the radioimmunoprecipitation assay buffer (RIPA buffer) (50 mM Tris, 150 mM NaCl, 1% Triton X-100, 0.1% SDS, and 1% Na-deoxycholate [pH 7.4]) with protease inhibitor and phosphatase inhibitor cocktails, and then subjected to SDS-PAGE. The proteins were transferred onto nitrocellulose membranes (GE Healthcare Life Science), blocked with Tris-buffered saline (TBS) containing 0.2% Tween 20 and 5% skim milk, incubated with the primary antibody, and then incubated with the HRP-labeled secondary antibody. Signals were detected using X-ray films. Blots were quantified using Image J software (NIH, Bethesda, MD, USA) and were normalized to β-actin as a loading control. The quantifications and statistics are presented in Appendix A. Original Western blot films can be found in Appendix A. 

### 2.6. Invasion Assay

For the invasion assay, Transwell chambers were coated with Matrigel (Corning, Tewksbury, MA, USA). Cells were seeded at 5 × 10^5^ cells per well in a coated Transwell chamber in 24-well plates. After treatment with CBD, cells were incubated at 37 °C for 48 h. Treated cells were stained using the Diff-Quik Stain Kit (Biochemical Sciences, Inc., Swedesboro, NJ, USA). Stained cells were mounted on slides and observed by microscopy.

### 2.7. Migration Assay (Wound Healing Assay)

For the wound healing assay, cells were seeded at 3 × 10^5^ cells per well in 12-well plates and incubated for 24 h. Cells were treated with CBD. After reaching almost 100% confluence, the cells were scratched using a pipette tip and then grown in a culture medium with 0.5% FBS for 24 h. Cell migration was monitored and captured using microscopy. The percentage of the recovered area was measured using ImageJ software.

### 2.8. Transient Transfection

HA-HIF-1α, HA-Ub, and pcDNA3-Src plasmids were purchased from Addgene (Watertown, MA, USA). Small interfering RNAs (siRNAs) targeting Src were purchased from Santa Cruz Biotechnology. Cells were transiently transfected with plasmids or siRNAs using Lipofectamine 2000 or RNA iMAX reagents (Invitrogen, Carlsbad, CA, USA) according to the manufacturer’s instructions.

### 2.9. Immunofluorescence (IF) Assay

Cells were fixed with 4% paraformaldehyde, permeabilized with 0.5% Triton X-100, blocked with 3% BSA solution, and incubated with anti-HIF-1α and anti-Src primary antibodies. Primary antibodies were visualized using an anti-mouse Alexa 488-conjugated secondary antibody or anti-rabbit Alexa 594 secondary antibody (Molecular Probes, Eugene, OR, USA), and the cells were stained with 4′,6-diamidino-2-phenylindole (Invitrogen). Finally, the cells were mounted and imaged using a confocal microscope (LSM700, Carl Zeiss, Oberkochen, Germany).

### 2.10. Real-Time Polymerase Chain Reaction (RT-PCR)

RNA was extracted using the TRIzol reagent (Life Technologies, Grand Island, NY, USA). Transcripts were amplified using a reverse transcriptase polymerase chain reaction kit (Life Technologies). Real-time PCR was performed on an Applied Biosystems 9700 real-time PCR system using gene-specific oligonucleotide primers for TaqMan probes (Applied Biosystems, Foster City, CA, USA). TaqMan probes were as follows: GAPDH (Hs99999905_m1), HIF-1α (Hs00153153_m1), and vascular endothelial growth factor (VEGF, Hs00900055_m1). For mRNA expression, gene expression was normalized t GAPDH.

### 2.11. Co-Immunoprecipitation (Co-IP)

Cells were washed with ice-cold phosphate-buffered saline (PBS) and incubated with 300 μL of a lysis buffer (1 mM phenylmethylsulfonyl fluoride, protease inhibitor, and phosphatase inhibitor; Cell Signaling Technology). Cells were harvested and cell debris was removed by centrifugation at 15,000 rpm for 5 min at 4 °C. Protein quantification was performed using a bicinchoninic acid assay (Thermo Scientific, Waltham, MA, USA). The supernatants were incubated with primary antibodies at 4 °C overnight. Protein G PLUS-agarose beads were added for 1 h at 4 °C. Immunoprecipitates were washed and separated by centrifugation at 15,000 rpm and heated at 100 °C with 2× sample buffer. The supernatants were then assessed using Western blotting. 

### 2.12. Enzyme-Linked Immunosorbent Assay (ELISA) for VEGF

To evaluate the amount of VEGF-A in the cells, the cells were centrifuged to remove cellular debris and assayed using the Human VEGF Quantikine ELISA Kit (DVE00; R&D Systems), according to the manufacturer’s instructions. 

### 2.13. HUVEC Tube-Formation Assay

For the angiogenesis assay, early passage (<P4) HUVECs were seeded at 3 × 10^4^ cells per well on Matrigel-coated 48-well plates. HUVECs were incubated with or without CBD treatment-conditioned medium. After 6 h, HUVEC tube formation was imaged under a phase-contrast inverted microscope, and the total tube lengths were quantified using the ImageJ program (NIH, Bethesda, MD, USA). HUVECs were grown with conditioned media with or without VEGF as a positive and negative control. The control image can be found in Appendix A.

### 2.14. Clonogenic Assay

Among the 500 or 1000 cells that were dissociated from the monolayer cells and the cultivated mammospheres, cells derived from the mammospheres were transfected with the HA-HIF-1α plasmid in both groups. After 14 days, the cells were transferred to each well of an adherent 6-well plate containing DMEM supplemented with 10% FBS and 1% streptomycin. After 14 days of culture, the dissociated cells were washed with PBS, fixed, and stained with a solution containing 20% ethanol, 3.7% formaldehyde, and 0.2% crystal violet. This process was repeated for three experiments.

### 2.15. Detection of CD44 and CD24 by Flow Cytometry (FACs) Analysis

Cell characterization was performed using a FACS Calibur flow cytometer (Becton Dickinson, Germany). The following monoclonal antibodies were used: Conjugated antibodies against anti-CD44 and CD24 (Invitrogen, Carlsbad, CA, USA). In brief, cultured cells were detached using trypsin EDTA (170,000 U trypsin/L; Lonza, Morristown, NJ, USA) and centrifuged at 1800 rpm for 10 min. The pellet was collected and washed once with PBS. For intracellular markers, the following additional steps were performed: Fixation was performed using paraformaldehyde (4%) in 1× PBS followed by vortexing, incubation for 30 min at room temperature, and washing with PBS. Permeabilization was performed using Triton (0.1, X-100) in 1× PBS, followed by incubation for 30 min at room temperature and then washing with PBS. Cells were re-suspended in blocking buffer to reduce non-specific binding (BSA (0.5%) in 1× PBS). They were then placed into Eppendorf tubes and vortexed before incubation on ice for 30 min. The cell pellet was re-suspended in 150 μL of the blocking buffer with 5 μL of the selected stain and incubated on ice for another 30 min. Afterwards, a wash step was performed and then the samples were incubated for 1 h in the dark at room temperature. The cells were finally washed with the blocking buffer and resuspended in PBS for analysis. The cells were then analyzed, and representative histograms were obtained using BD CellQuest™ Pro software (Becton Dickinson, Germany). 

### 2.16. Statistical Analysis

Each assay was performed in triplicate and independently repeated at least three times. Statistical analyses were performed using GraphPad InStat 6 software (GraphPad, Inc., La Jolla, CA, USA). Statistical significance was defined as *p* values < 0.05 (*, **, and *** indicate *p* < 0.05, *p* < 0.01, and *p* < 0.001, respectively).

## 3. Results

### 3.1. CBD Reduces Proliferation, Invasion, and Migration of Breast Cancer Cells

To determine whether CBD influences the proliferation of breast cancer cells, multiple breast cancer cell lines, including MCF7, T47D, SK-BR-3, and MDA-MB-231 cells, and normal breast cell lines (MCF10A) were cultured with different concentrations (0–7 μM) of CBD (Figure 1A). It was demonstrated that apoptosis does not occur at a specific CBD concentration (2 μM). Western blotting showed that the expression of Slug, which is an endothelial–mesenchymal transition (EMT)-related protein, was decreased by CBD (2 µM) treatment (Figure 1B). We found that CBD can affect EMT. The invasion of both MCF7 and MDA-MB-231 cells was significantly reduced in the CBD treatment group (TG) (*p* < 0.001) compared to the control group (CG) (Figure 1C). In addition, using the wound-healing assay, it was demonstrated that the migration of MCF7 cells induced by CBD was reduced compared to that of the CG (*p* < 0.01) and MDA-MB-231 cells (not significantly, NS) (Figure 1D).

### 3.2. CBD Decreases HIF-1α by Upregulating the Ubiquitination of HIF-1α

It was demonstrated that HIF-1α expression was increased under hypoxia (1% O_2_) and decreased in the TG in all breast cancer cell lines for 24 h (Figure 2A,B). To validate these results, we exposed MCF7 cells to cobalt chloride (CoCl_2_), a hypoxia-inducing agent, and transfected them with HA-HIF-1α plasmid under normoxic conditions in both groups for 24 h. The expression of HIF-1α in MCF7 cells was increased significantly upon exposure to CoCl_2_ and transfection with HA-HIF-1α plasmid in the CG; however, it was decreased in the TG (Figure 2C,D). In addition, immunofluorescence analysis demonstrated that the immunoreactivity of HIF-1α was decreased in MCF7 cells transfected with HA-HIF-1α plasmid in the TG (Figure 2E). However, there was no significant difference in the mRNA expression levels of HIF-1α as shown by RT-PCR (Figure 2F). Thus, it was expected that the expression of HIF-1α in the CBD TG might be regulated by post-translational modification and that CBD could play a role in the degradation of HIF-1α protein. The effect of MG132, a proteasome inhibitor, on the degradation of HIF-1α and the exploratory interaction of HIF-1α with ubiquitin was determined by Western blotting and co-immunoprecipitation assays. It was revealed that MG132 inhibited the degradation of HIF-1α in the TG (Figure 2G); however, CBD promoted it by ubiquitination following the HIF-1α–ubiquitin interaction (Figure 2H).

### 3.3. CBD Modulates the Expression Pattern of Kinases in MCF7 Cells

CBD significantly reduced Src expression levels in all breast cancer cell lines (Figure 3A). The average intensity of Src in MCF7 cells was measured with or without CBD treatment in an immunofluorescence assay. It was demonstrated that the Src signal intensity could hardly be detected in CBD-treated cells compared to the control cells (Figure 3B). The reduction of Src expression in the CBD TG when pcDNA3-Src and Src siRNA were used demonstrated the inhibition of Src by CBD (Figure 3C,D). In addition, the inhibition of Src by CBD increased the expression of VHL and decreased that of HIF-1α. The interaction between Src, VHL, and HIF-1α in the CBD TG was determined by a co-immunoprecipitation assay (Figure 3E). It was thus demonstrated that CBD promoted the inverse interaction between Src/VHL and VHL/HIF-1α.

### 3.4. CBD Inhibits Angiogenesis in Breast Cancer

It is known that the expression of VEGF in hypoxia is mainly regulated by the activation of HIF-1α, so we investigated whether CBD could affect the mRNA expression of VEGF and the amount of secreted VEGF by inhibiting the activation of HIF-1α. The expression of VEGF was higher in hypoxia than in normoxia. However, there was a decreasing trend in the TG in both oxygen status groups in MCF7 and MDA-MB-231 cells (Figure 4A). Similarly, the expression of VEGF in hypoxia increased in the CG of both cells but was significantly decreased in the TG (*p* < 0.001) (Figure 4B). It was also demonstrated that the angiogenesis of all four breast cancer cell lines was significantly suppressed in the TG by the HUVEC tube formation assay (Figure 4C,D). These observations demonstrate that CBD contributes to the suppression of angiogenic ability in breast cancer cells.

### 3.5. CBD Reduces Stemness of Breast Cancer

To investigate the effect of CBD on the stemness of mammospheres, the development of mammospheres derived from MCF7 cells and their stemness could be identified by flow cytometry and clonogenic assays. It was demonstrated that mammospheres expressed higher levels of CD44 and CD24 than cultured monolayer cells (Figure 5A,B). It was also found that CBD inhibited the formation and growth of mammospheres (Figure 5C–E). The expression of stem cell markers NANOG, SOX2, CD133, and ALDH1A1 was evaluated by Western blotting in mammospheres in both groups. The expression levels of all stem cell markers were significantly increased in the mammospheres in the CG compared to monolayer cells but decreased more in the mammospheres in the TG than in the CG (Figure 5F). By investigating the expression of stem cell markers and HIF-1α/Src/VHL in mammospheres in both groups, we found that the expression of stem cell markers and HIF-1α was decreased through Src/VHL/HIF-1α signaling by CBD in the mammospheres, as in monolayer MCF7 cells (Figure 5F,G).

## 4. Discussion

It has been established that CBD could be a potential anti-cancer drug candidate, which has functions linked to various signaling pathways [18]. Although we focused on angiogenesis related to HIF-1α in CBD treatment in our study, we demonstrated that CBD inhibits the proliferation, migration, and invasion of breast cancer cells. CBD reduced the proliferation rate at higher CBD concentrations in all breast cancer cell lines, but not in the normal breast cells. It was reported that the viability of MCF7 cells at a specific concentration of CBD (10 μM) was reduced to 90% by CBD and this occurred through interaction with the CB1 receptor [19]. Although it is known that the relationship between CB receptors and the anti-cancer effect of CBD has been uncertain, it was reported that CBD downregulated the CB1 receptor level through the low expression of CNR1 and could reduce the viability of MCF7 cells [19]. CBD also downregulated the expression of EMT markers such as Slug and Vimentin. CBD might cause breast cancer cells to be less aggressive by mesenchymal reversion to an epithelial phenotype via IL-1β/IL-1R/β-catenin, E-cadherin/β-catenin complex relocalization, and other protein markers of malignancy [19]. Thus, it could be understood that CBD can inhibit the aggressiveness of breast cancer and be sensitized to more aggressive breast cancer cells by regulating a variety of signaling pathways and immunologic factors related to malignancy as well as Src/VHL/HIF-1α signaling. HIFs are heterodimeric transcription factors that consist of an O_2_-regulated HIF-α subunit (HIF-1α, HIF-2α, or HIF-3α), which remains inactive through hydroxylation in normoxia [20]. HIF-1α was shown to regulate the transcription of target genes such as VEGF, resulting in the activation of hypoxia adaptive pathways such as the heterodimeric complex, which are formed by dimerizing HIF-α and HIF-β subunits, binding to the hypoxia-responsive elements (HRE) of target genes [21,22]. Given that VHL is a critical E3 ubiquitin ligase that recognizes prolyl hydroxylase domain (PHD)-mediated hydroxylation of proline residues in normoxia, it is known that the function of VHL for oxygen-dependent HIF-1α protein degradation could be maintained even under hypoxia, as our study has shown [23,24,25,26]. It has been reported that a substantial proportion of HIF-1α in the majority of tumors under hypoxic conditions are proline-hydroxylated, and the degradation of proline-hydroxylated HIF-1α by VHL might be rate limiting because PHD could not be completely blocked or HRE could be attacked in hypoxia [27]. In addition, Src was shown to have a significant impact on VHL/HIF-1α interaction. Src belongs to a family of nonreceptor tyrosine kinase proteins that can contribute to cancer development and progression when mutated or overexpressed [28] and has been implicated as an important signaling component in hypoxia-induced upregulation of VEGF [29,30]. Src has been reported to decrease or destabilize the VHL protein through tyrosine residue 185 phosphorylation, leading to the ubiquitination of VHL by the E2 ubiquitin-conjugating enzyme and the degradation of VHL via the 26S proteasome [28]. Src also induces NADPH oxidase/Rac-dependent ROS generation, which can impose restrictions on hydroxylation-dependent VHL recruitment of HIF-1α [31]. However, it was demonstrated that Src attenuation by CBD caused the stabilization of VHL as well as the downregulation of HIF-1α and VEGF, followed by the inhibition of tumor angiogenesis under hypoxic conditions (Figure 6). To the best of our knowledge, this is the first study to show that the inhibition of Src activity caused the VHL level to increase, followed by decreasing HIF-1α and VEGF activation in hypoxia, and CBD could prevent angiogenesis in breast tumors through Src/VHL/HIF-1α signaling. In addition to triggering HIF-dependent transactivation of multiple genes required for cancer invasion and metastasis, HIF-1α also plays key roles in the specification and maintenance of CSCs, whose numbers are increased by intratumoral hypoxia [12,17,32]. Mammospheres or breast cancer cells in hypoxia have higher expression of the KLF4, NANOG, OCT4, and SOX2 genes and CD44 and CD24, which encode pluripotency factors that are required for the maintenance of cancer stem cells, embryonic stem cells, and induced pluripotent stem cells [33,34,35,36]. However, there is controversy regarding the extent to which the stemness is enriched according to the degree of expression of CD44 and CD24 and its relation to clinical outcomes or molecular subtypes [37,38,39,40,41,42]. It was demonstrated that CBD can inhibit the abilities of BCSCs by decreasing the expression of CSC markers such as ALDH1A1, CD133, NANOG, and SOX2 in mammospheres. It was also revealed that CBD can decrease Src and HIF-1α levels and inhibit the growth and proliferation of mammospheres.

## 5. Conclusions

CBD inhibits Src activity, which decreases HIF-1α through the degradation of VHL recruitment and the direct reduction of HIF-1α protein synthesis in mammospheres as well as breast cancer cells. Therefore, CBD can attenuate Src/VHL/HIF-1α signaling and inhibit angiogenesis, followed by inhibiting the growth and invasion of breast cancer and cancer stem-like properties. Further studies are needed to demonstrate the potential of CBD as a treatment for refractory or recurrent breast cancer as well as primary breast cancer.

## Figures and Tables

**Figure 1 cancers-13-05667-f001:**
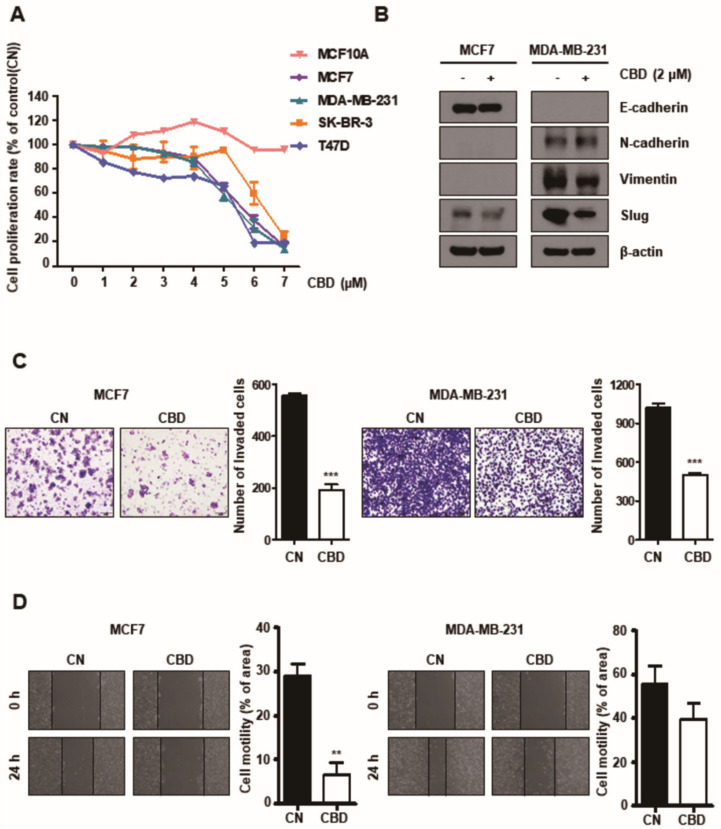
Cannabidiol (CBD) reduces migration and invasion in breast cancer cell lines. (**A**) Breast cancer cell lines and normal cell lines were treated with different doses of CBD for 24 h and assessed using the WST assay. (**B**) Expression of EMT-related proteins was evaluated using Western blotting. β-actin was used as a loading control. (**C**) Cell invasiveness was measured using a Transwell assay with Matrigel. The invaded cells were stained and visualized using microscopy. *** *p* < 0.001 (**D**) Cell migration ability was measured by a wound-healing assay. Cells were scratched with pipette tips and then incubated with 0.5% FBS in conditioned media with or without CBD (2 μM). The percentage of recovered area was quantified using ImageJ. **, *p* < 0.01 CBD; cannabidiol, CN; control.

**Figure 2 cancers-13-05667-f002:**
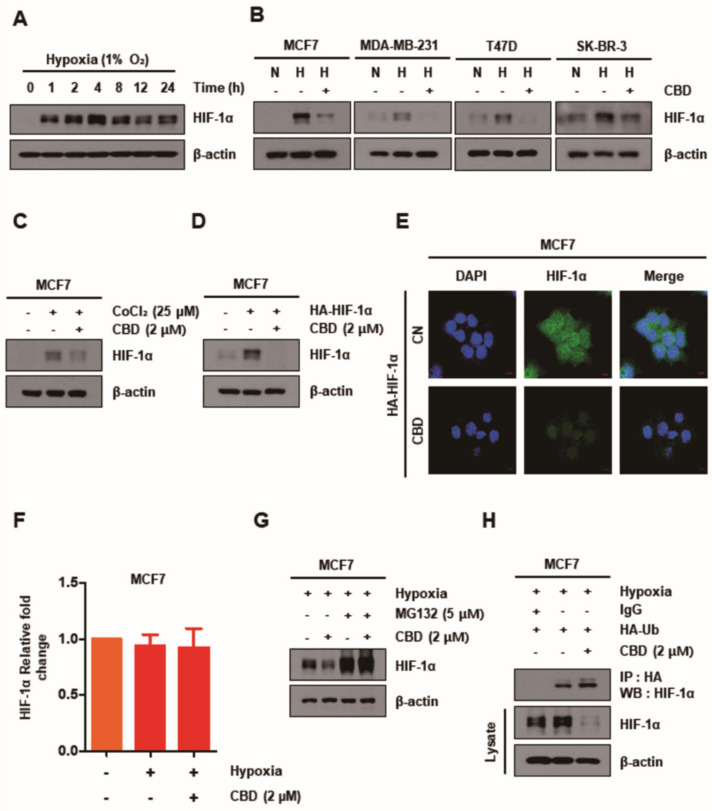
CBD decreases HIF-1α by post-transcriptional levels of HIF-1α regulating the ubiquitin proteasome (UPS) system. (**A**) MCF7 cells were treated with CBD (2 μM) for different time periods in hypoxia conditions. The expression of HIF-1α detected by Western blotting was significantly increased after 4 h. (**B**) Breast cancer cell lines were treated with CBD (2 μM) for 24 h. The protein level of HIF-1α was detected by Western blotting. (**C**,**D**) MCF7 cells were chemically induced with CoCl₂ (25 µM) or transfected with HA-HIF-1α plasmid. Then, cells were treated with CBD (2 μM) for 24 h in normoxia conditions. The protein level of HIF-1α was detected by Western blotting. (**E**) MCF7 cells were transfected with HA-HIF-1α plasmid and then treated with CBD (2 μM) in normoxic conditions. Immunofluorescence images showed the expression of HIF-1α. (**F**) The mRNA level of HIF-1α was measured by qRT-PCR. (**G**) MCF7 cells were pretreated with proteasome inhibitor (MG132) (5 µM). After 1 h pretreatment, the cells were treated with CBD (2 μM) for 24 h. The protein level of HIF-1α was detected by Western blotting. β-actin was used as a loading control. (**H**) MCF7 cells were transfected with HA-Ub plasmid and then treated with CBD (2 μM) for 24 h. HIF-1α was isolated by immunoprecipitation with IgG and anti-HA antibody, followed by immunoblotting for HIF-1α; total HIF-1α and β-actin protein levels were detected by Western blotting. CBD; cannabidiol, CN; control, N; normoxia, H; hypoxia.

**Figure 3 cancers-13-05667-f003:**
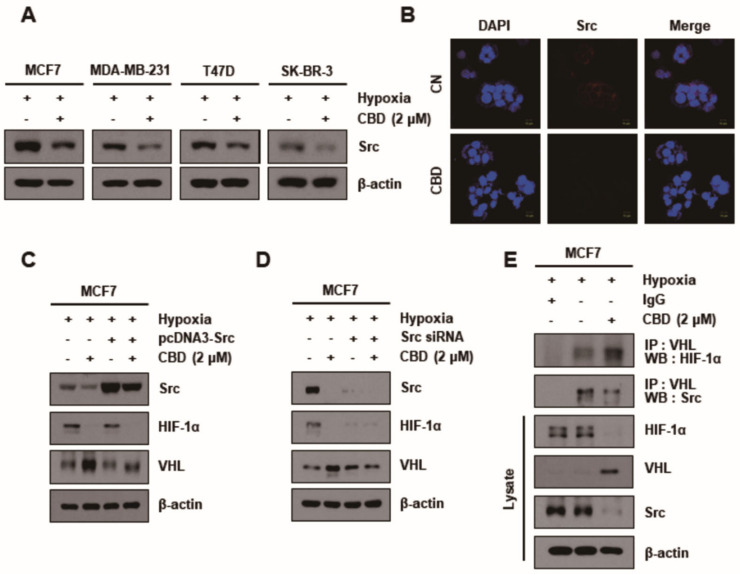
Src regulates the HIF-1α expression via ubiquitination of VHL. (**A**) MCF7 cells were treated with or without CBD (2 μM) in hypoxia conditions. The expression level of Src detected by Western blotting. β-actin was used as a loading control. (**B**) Immunofluorescence images showed the expression of Src in MCF7 cell lines. (**C**) MCF7 cells were transiently transfected with pcDNA3-Src plasmid or (**D**) Src siRNA and then treated in both groups under hypoxia. The cells were then lysed, and Western blotting was performed for Src, HIF-1α, VHL, and β-actin. β-actin was used as a loading control. (**E**) The interaction between VHL and HIF-1α or Src was assessed by co-immunoprecipitation assay. Cells were treated with CBD (2 μM) in hypoxia conditions. Cell lysates were immunoprecipitated with the IgG and anti-VHL antibody, followed by immunoblotting for HIF-1α, Src, total HIF-1α, VHL, and β-actin. β-actin was used as a loading control. CBD; cannabidiol, CN; control.

**Figure 4 cancers-13-05667-f004:**
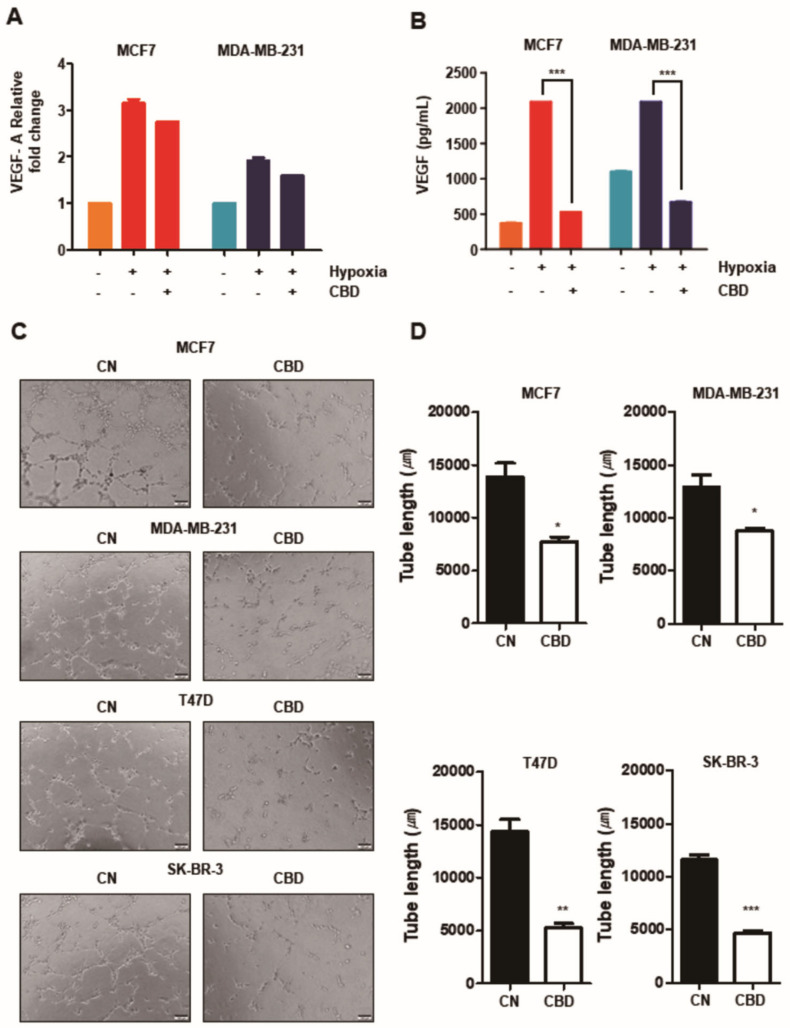
CBD inhibits angiogenic potential in breast cancer cell lines. (**A**) MCF7 and MDA-MB-231 cells were treated with CBD (2 μM) in hypoxia. The total mRNA level of VEGF was analyzed by performing qRT-PCR. (**B**) The amount of secreted VEGF protein in MCF7 and MDA-MB-231 cells was analyzed by ELISA Kit. ***, *p* < 0.001 (**C**) The angiogenic potential was determined by HUVEC tube-formation assay. Conditioned media was collected from the breast cancer cell lines in hypoxia. Early-passage HUVECs were grown on matrigel-coated 48-well plates with conditioned media for 6 h. Capillary tube formation in each well was captured using a light microscope. (**D**) Total tube length was quantified using the ImageJ program. *, **, and *** indicate *p* < 0.05, *p* < 0.01, and *p* < 0.001 CBD; cannabidiol, CN; control.

**Figure 5 cancers-13-05667-f005:**
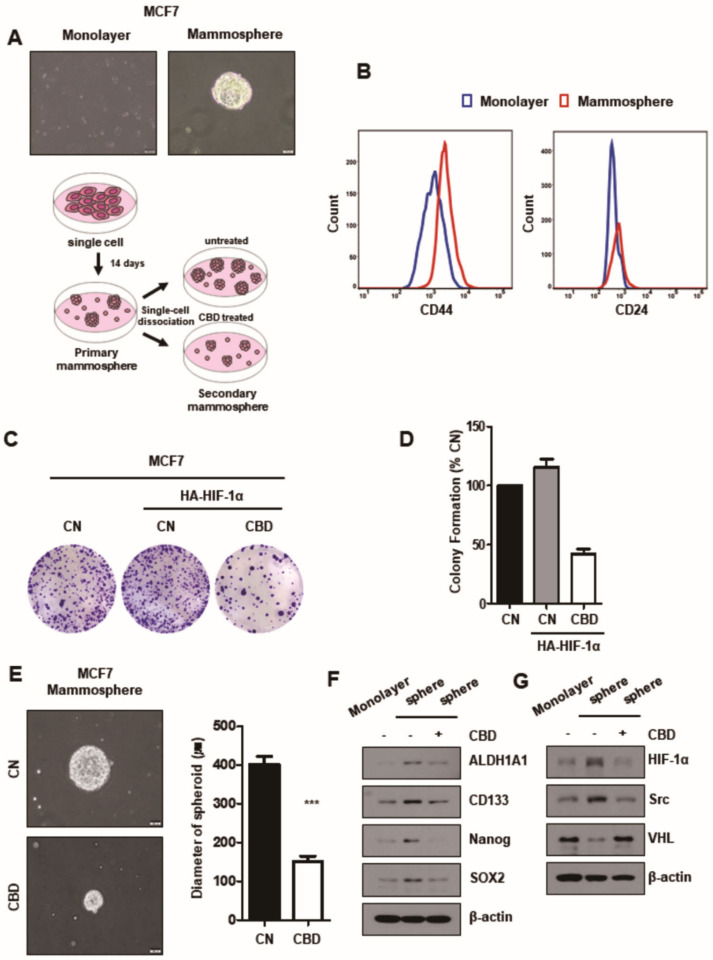
CBD reduces stemness of MCF7 breast cancer cells. (**A**) The sphere-forming cells were cultured in ultra-low coated-dishes for two weeks. Then, sphere-forming and monolayer cells were captured using a bright-field microscope. The scheme of the sphere-forming assay. (**B**) The monolayer cells and sphere-formed MCF7 cells were stained with breast cancer stem cell markers (CD44 or CD24) and analyzed by flow cytometry. (**C**) MCF7 cells were transfected with HA-HIF-1α plasmid and treated with 2 µM CBD. Cells were seeded on 6-well plates. After two weeks, the colonies were stained with crystal violet dye. (**D**) The number of colonies was counted. Colony numbers were identified in three experiments. (**E**) The sphere-forming cells were seeded in new ultra-low coated-dishes with or without CBD (2 μM) for two weeks. Then, the secondary sphere-forming cells were captured using a bright-field microscope. The diameters of the spheroids in each dish were measured. ***, *p* < 0.001 (**F**) The stem cell-related proteins and (**G**) the protein levels of HIF-1α, Src, and VHL were measured by Western blotting in both groups in secondary sphere-forming cells. β-actin was used as a loading control. CBD; cannabidiol, CN; control.

**Figure 6 cancers-13-05667-f006:**
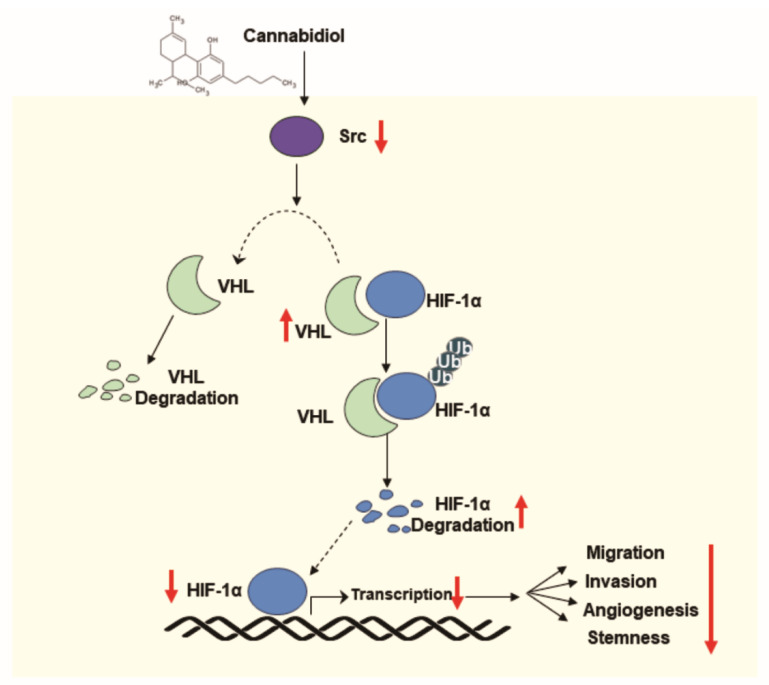
The scheme of cannabidiol (CBD) suppression of angiogenesis and stemness of breast cancer by downregulation of HIF-1α.

## Data Availability

The data presented in this study are available in this article (and Appendix A).

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
