# Peer review of "Cannabidiol Suppresses Angiogenesis and Stemness of Breast Cancer Cells by Downregulation of Hypoxia-Inducible Factors-1α"

_cancers, 2021, doi:10.3390/cancers13225667_

Round 1

Reviewer 1 Report

MJ Job et al explored the cytotoxic, anti-invasive, anti-migration, anti-angiogenic and stemness suppressive effects of CBD, through Src/VHL/HIF-1α signalling, in in-vitro models of breast cancer. The manuscript is professionally written and here are my comments or recommendations to be considered by the authors:

Critiques

  • A better discussion of the results will greatly enhance the manuscript. The current discussion is fragmented and is covering the previous relevant literature well, but not the present results. The current format may add up to the displayed thin introduction but not to the discussion section.
  • Please visualise the western blotting results as bar or box plots showing the stats significance among experimental groups (for example in figure 2C, and 2G….etc.).

Minor

  • P4 L147 please mention the heating up temperature with the 2X sample buffer.
  • Figure 5 legend “CBD reduces stemness of breast cancer cell lines” should be amended to MCF7 breast cancer cells as other cell lines are not studied.
  • Relevant references are to be discussed including recent label-free quantification proteomic study of CBD against MCF7 cells where HIF-1α was not differentially expressed compared to control in agreement with the current RT-PCR results.https://doi.org/10.3390/ijms221810103
  • P15, L 361 “Solid tumour” is not an accurate generalisation as not studied.
  • CBD concentration is missing for example in Figure 1C-D, Figure 2B or the corresponding legends, please ensure it is mentioned for different assays or mention “2uM” throughout all studies if it was the case.
  • Please add the cell line name used in figure 3B and within the figure legends as well.
  • Some higher resolution subfigures are needed.
  • Typos and grammar mistakes for examples:

P8, L236 please amend “cell” to cells

P11, L283 please amend “was” to were

P10, L272 and 273 please amend VEGF-A to VEGF or mention whats meant by VEGF-A if intended

Author Response

  • A better discussion of the results will greatly enhance the manuscript. The current discussion is fragmented and is covering the previous relevant literature well, but not the present results. The current format may add up to the displayed thin introduction but not to the discussion section.

We appreciate the reviewer’s critiques. According to the reviewer comment, we added and revised the ‘introduction’ and ‘discussion’ section.

  • Please visualise the western blotting results as bar or box plots showing the stats significance among experimental groups (for example in figure 2C, and 2G….etc.).

Thank you for your keen observation. As recommended, we now revised the western blot data in the Figure. Moreover, we quantified the data with Image J program and presented as bar graph showing the statistic. We revised in accordingly in the ‘Supplementary Figure’ section.

Minor

  • P4 L147 please mention the heating up temperature with the 2X sample buffer.

Thank you for your comments. We added heating temperature accordingly in revised version of manuscript.

  • Figure 5 legend “CBD reduces stemness of breast cancer cell lines” should be amended to MCF7 breast cancer cells as other cell lines are not studied.

We thank the reviewer for pointing out. We revised accordingly in the revised manuscript.

  • Relevant references are to be discussed including recent label-free quantification proteomic study of CBD against MCF7 cells where HIF-1α was not differentially expressed compared to control in agreement with the current RT-PCR results. https://doi.org/10.3390/ijms221810103

Thank you for your precious comments. In our study, we investigated the expression of both HIF-1α mRNA and protein in the MCF7 breast cancer cells with CBD treatment. We found that there was no difference in HIF-1α mRNA expression in hypoxia and CBD treatment. However, the expression of HIF-1α protein was markedly induced in hypoxia and decreased in CBD treatment in MCF 7 cells. This revealed that the regulation of HIF-1α is post-translational level of protein turnover. In fact, the canonical regulation of HIF-1α protein stability and activity involves post-translational modifications (Ref a and b). In this regard, we provides that CBD decreases the Src/ HIF-1α by regulating ubiquitin proteasome system in post-translational level.

Ref a and b:

  1. Adam Albanese, International Journal of Molecular Sciences: The Role of Hypoxia-Inducible Factor Post-Translational Modifications in Regulating Its Localisation, Stability, and Activity : 2021.22,268
  2. Crosson, Investigate ophthalmology & visual science: Time Course of Hypoxia Inducible Factor-1 alpha (HIF-1) mRNA and Protein Levels in Rat Retina, 2005 May, 46(13)
  • P15, L 361 “Solid tumour” is not an accurate generalisation as not studied.

Thank you for your comments. We revised accordingly in the revised manuscript

  • CBD concentration is missing for example in Figure 1C-D, Figure 2B or the corresponding legends, please ensure it is mentioned for different assays or mention “2uM” throughout all studies if it was the case.

Accepting the reviewer comment, we revised the CBD concentration in legends and figures.

  • Please add the cell line name used in figure 3B and within the figure legends as well.

We appreciate the reviewer to point out. We added the cell line name (MCF7) in figure 3B and figure legends.

  • Some higher resolution subfigures are needed.

We are terribly sorry and revised higher resolution figure and added the quantification graph of western blot.

  • Typos and grammar mistakes for examples:

P8, L236 please amend “cell” to cells

P11, L283 please amend “was” to were

P10, L272 and 273 please amend VEGF-A to VEGF or mention whats meant by VEGF-A if intended

Thank you for your comments. We revised typos and grammar mistakes including above examples and now reported in revised version of manuscript.

We look forward to seeing our manuscript in your journal.

Sincerely,

Woo Young Kim

Reviewer 2 Report

Cannabidiol or CBD has been previously shown to inhibit migration, invasion, and metastasis in aggressive breast cancer in vivo and in vitro (Breast Cancer Res Treat. 2011 Aug;129(1):37-47, Br J Pharmacol . 2014 Oct;171(19):4464-77, Mol Oncol . 2015 Apr;9(4):906-19, J Pharmacol Exp Ther . 2006 Sep;318(3):1375-87), therefore additional elucidation of CBD effect on breast cancer is a crucial manner when looking at CBD potential as a therapy in breast cancer.  

Even though this is a well-organized, thorough, and methodological research, I believe this manuscript will highly benefit from a few additional experiments straightening the main message.    

  1. Since CB2receptor has also been found to be overexpressed in HER2+ breast cancers (J Natl Cancer Inst . 2015 Apr 8;107(6):djv077) and was seen as a pivotal regulator of HER2 pro-oncogenic signaling. Although the introduction states that any cannabinoid receptor does not mediate the CBD effect, a demonstration of this effect will strongly contribute to this claim. For example, performing some of these assays in the presence of CNR-chemical blockers or the use of CNR KD/KO cell-lines  

  1. Likewise, It was shown that CBD exerts its anti-proliferative effects on breast cancer cells through various mechanisms, including apoptosis, autophagy, and cell cycle arrest; therefore, the demonstration of other mechanisms besides apoptosis is needed.

  1. Figure 3- Panel A: Not clear by the legend if the presented Scr band represents total or phosphor-Scr. It is stated in the figure legends that it was probed with Human Phospho Kinase Array Kit. Please clarify. In addition, if this is Phospho Scr, also use total unphosphorylated Scr as a control. Panel B- the text is B) The protein level of Src was detected by western blotting in both groups in normoxia or hypoxia conditions". The actual figure present IF stain of total Scr. Enhancing stain quality or intensity is needed since Red stain for Scr is hardly seen in the control.

  1. The angiogenic potential using tube formation assay should also be performed with a control medium with neutralization of VEGF. Figure-4C better clarifies what the control is conditional medium.

  1. "Thus, it is expected that CBD can be used to target breast cancer refractory to hormonal therapy, which is known to be associated with the stemness of breast cancer cells, as it decreases breast cancer stem cell-like properties and inhibits the growth and invasion of mammospheres through 377 Src/VHL/HIF-1α signaling."- This statement should be moderated without any in vivo animal model included in the study or any clinical evidence to support it. Mainly when CBD affects on the cancer microenvironment and its potent immunosuppression properties, do not consider or discuss   

Author Response

We appreciate the helpful comments from the reviewers. It has helped to improve the manuscript, and to clarify some points that might cause confusion with the readers. We have made revisions (as marked in red font in the revised version of manuscript) according to the reviewer’s comments.

Please see figures on the attachment file.

Reviewer 2

Cannabidiol or CBD has been previously shown to inhibit migration, invasion, and metastasis in aggressive breast cancer in vivo and in vitro (Breast Cancer Res Treat. 2011 Aug;129(1):37-47, Br J Pharmacol . 2014 Oct;171(19):4464-77, Mol Oncol . 2015 Apr;9(4):906-19, J Pharmacol Exp Ther . 2006 Sep;318(3):1375-87), therefore additional elucidation of CBD effect on breast cancer is a crucial manner when looking at CBD potential as a therapy in breast cancer.  

Even though this is a well-organized, thorough, and methodological research, I believe this manuscript will highly benefit from a few additional experiments straightening the main message.    

We would like to express our sincere gratitude to the respected reviewer for appreciation to our research.

#1

Since CB2receptor has also been found to be overexpressed in HER2+ breast cancers (J Natl Cancer Inst . 2015 Apr 8;107(6):djv077) and was seen as a pivotal regulator of HER2 pro-oncogenic signaling. Although the introduction states that any cannabinoid receptor does not mediate the CBD effect, a demonstration of this effect will strongly contribute to this claim. For example, performing some of these assays in the presence of CNR-chemical blockers or the use of CNR KD/KO cell-lines  

We thank the reviewer for pointing this out. According to the reviewer’s suggestion, we investigated the expression of CB receptors upon CBD treatment in HER2+ breast cancer cells (MCF7) and HER2+ breast cancer cells (SK-BR-3).

As a result, there was no difference in the expression of CB1 in both cell lines upon CBD treatment. The expression of CB2 was increased in the HER2+ breast cancer cells (SK-BR-3) compared to the HER2- breast cancer cells (MCF7), but CBD treatment did not affect the expression of CB2. In this regard, we considered that cannabinoid receptors do not mediate CBD effects in this study.

Ref a:

  1. Daniel J, Cancer Metastatics Rev: Cannabinoids, Endocannabinoids and Cancer. 2011 Dec;30(3-4)

#2

Likewise, It was shown that CBD exerts its anti-proliferative effects on breast cancer cells through various mechanisms, including apoptosis, autophagy, and cell cycle arrest; therefore, the demonstration of other mechanisms besides apoptosis is needed.

We thank the reviewer for this suggestion. We have identified a variety of mechanisms including apoptosis, autophagy to determine the effects of CBD on breast cancer. We confirmed that CBD treatment affects dose-dependent apoptosis in MCF 7 breast cancer cells through western blot and flow cytometry analysis.

In addition, CBD treatment increased LC3 I/II and p62, which is known as autophagy marker, in MCF7 breast cancer cells.  

We performed flow cytometry analysis to demonstrate the cell cycle in CBD treatment. As shown in data, CBD slightly induced sub G1 phase and increased S phase in MCF7 breast cancer cells. G2M phase was decreased with CBD treatment in cells. However, to demonstrate the cell cycle on CBD effect would be needed with further investigation as other research.

In this study, we focused on the invasion and angiogenesis via reducing Src/HIF1α signaling pathway in breast cancer cells. Although CBD has been shown to affect various mechanisms such as apoptosis, autophagy and cell cycle arrest, elucidating the detailed mechanisms involved is one research topic and will be covered in another further study.

#3

Figure 3- Panel A: Not clear by the legend if the presented Scr band represents total or phosphor-Scr. It is stated in the figure legends that it was probed with Human Phospho Kinase Array Kit. Please clarify. In addition, if this is Phospho Scr, also use total unphosphorylated Scr as a control. Panel B- the text is B) The protein level of Src was detected by western blotting in both groups in normoxia or hypoxia conditions". The actual figure present IF stain of total Scr. Enhancing stain quality or intensity is needed since Red stain for Scr is hardly seen in the control.

We are sorry to confuse you and thank you for your keen observation. In this study, the presented band represents total Src.

We confirmed that various kinds of phosphorylated protein expression through Human Phospho Kinase Array Kit upon CBD treatment, and Src protein was considered as a target based on the reference b.

Ref b:

  1. b) Mary T.-H.Chou, Genes & Cancer : The von Hippel-Lindau Tumor Suppressor Protein Is Destabilized by Src : Implications for Tumor Angiogenesis and Progression, 2010 ;1(3) 225-238

As shown in Figure 3A in our manuscript, it was confirmed through western blot that the expression of total Src was also decreased in CBD treated breast cancer cells.

As suggested by the reviewer, we changed the figure present IF of total Src and these data are now reported in our revised figure 3B in manuscript.

#4

The angiogenic potential using tube formation assay should also be performed with a control medium with neutralization of VEGF. Figure-4C better clarifies what the control is conditional medium.

We appreciate the reviewer’s comment and agree with the reviewer. In consideration of the reviewer’s point, a negative control without VEGF and a positive control treated with 25 ng/ml of VEGF were added to compare tube formation in the presence of VEGF. These data are now reported in our revised manuscript and supplementary figure S14. 

#5

"Thus, it is expected that CBD can be used to target breast cancer refractory to hormonal therapy, which is known to be associated with the stemness of breast cancer cells, as it decreases breast cancer stem cell-like properties and inhibits the growth and invasion of mammospheres through 377 Src/VHL/HIF-1α signaling."- This statement should be moderated without any in vivo animal model included in the study or any clinical evidence to support it. Mainly when CBD affects on the cancer microenvironment and its potent immunosuppression properties, do not consider or discuss   

We thank the reviewer for this suggestion. We accepted the reviewer’s opinion, the manuscript was revised and reported as a revised version of manuscript.

We look forward to seeing our manuscript in your journal.

Sincerely,

Woo Young Kim

Round 2

Reviewer 2 Report

The authors made many well-appreciated efforts to address all issues raised by both reviewers in an exquisite set of experiments and text modifications.

The manuscript is now ready for publication!